# Bandit Algorithm Driven by a Classical Random Walk and a Quantum Walk

**DOI:** 10.3390/e25060843

**Published:** 2023-05-25

**Authors:** Tomoki Yamagami, Etsuo Segawa, Takatomo Mihana, André Röhm, Ryoichi Horisaki, Makoto Naruse

**Affiliations:** 1Department of Information Physics and Computing, Graduate School of Information Science and Technology, The University of Tokyo, 7-3-1 Hongo, Bunkyo, Tokyo 113-8656, Japan; takatomo_mihana@ipc.i.u-tokyo.ac.jp (T.M.); roehm@g.ecc.u-tokyo.ac.jp (A.R.); horisaki@g.ecc.u-tokyo.ac.jp (R.H.); makoto_naruse@ipc.i.u-tokyo.ac.jp (M.N.); 2Graduate School of Environment and Information Sciences, Yokohama National University, 79-1 Tokiwadai, Hodogaya, Yokohama 240-8501, Kanagawa, Japan; segawa-etsuo-tb@ynu.ac.jp

**Keywords:** random walk, quantum walk, bandit algorithm, exploration–exploitation trade-off, decision-making

## Abstract

Quantum walks (QWs) have a property that classical random walks (RWs) do not possess—the coexistence of linear spreading and localization—and this property is utilized to implement various kinds of applications. This paper proposes RW- and QW-based algorithms for multi-armed-bandit (MAB) problems. We show that, under some settings, the QW-based model realizes higher performance than the corresponding RW-based one by associating the two operations that make MAB problems difficult—exploration and exploitation—with these two behaviors of QWs.

## 1. Introduction

Random walk (RW) is one of the most ubiquitous stochastic processes and is employed for both mathematical analyses and applications, such as describing real-world phenomena and constructing various algorithms. Meanwhile, along with the increasing interest in quantum mechanics from both theoretical and applied perspectives, the quantum counterpart of an RW, known as a quantum walk (QW), is also attracting attention [1,2,3,4]. A QW includes the effects of quantum superposition or time evolution. In classical RWs, a random walker (RWer) selects in which direction to go probabilistically at each time step, and thus, one can track where the RWer is at any time step. On the other hand, in QWs, one cannot tell where a quantum walker (QWer) exists during the time evolution, and the location is determined only after conducting the measurement.

QWs have a property that classical RWs do not possess: the coexistence of *linear spreading* and *localization* [5,6]. As a result, QWs show probability distributions that are totally different from those of random walks, which weakly converge to normal distributions. The former behavior, linear spreading, means that the standard deviation of the probability distribution of measurement of quantum walkers (QWers) grows in proportion to the run time *t*. In the case of discrete-time RWs on a one-dimensional lattice Z, denoting the random variable of the position where a walker is measured at time $t\in N_{0}=N\cup \left\{0\right\}$ by $X_{t}^{\left(\mathrm{RW}\right)}$, then the standard deviation is $D\left[X_{t}^{\left(\mathrm{RW}\right)}\right]=O\left(\sqrt{t}\right)$. On the other hand, in discrete-time QWs on Z, the standard deviation of a walker’s position at time *t* is $D\left[X_{t}^{\left(\mathrm{QW}\right)}\right]=O\left(t\right)$, and thus, discrete-time QWs outperform RWs in terms of the propagation velocity [7]. The latter behavior, localization, implies that the probability is distributed at a particular position no matter how long the walk runs. In the classical RWs, the probability distribution becomes flat despite keeping a bell-shaped curve; that is, localization is not observed.

QWs were first introduced in the field of quantum information theory [7,8,9]. The idea of weak convergence, which is frequently used in probability theory, was introduced to show the properties of QWs [10,11], and since then, quantum walks have been actively studied from both fundamental and applied perspectives. In fundamental fields, there have been many attempts to analyze these evolution models mathematically [6,12,13,14,15,16,17,18,19,20,21,22] due to varying behavior of QWs depending on the conditions or settings of time and space. In applied fields, their unique behavior is useful for implementing quantum structures or quantum analogs of existing models; therefore, various QW-based models have been considered for subjects such as time-series analysis [23], topological insulators [24,25], radioactive waste reduction [26,27], and optics [28,29]. In addition, the contribution to quantum information technology is becoming more prominent these days. QWs have been applied not only to the principle of technologies such as quantum search, quantum teleportation [30,31], and quantum key distribution [32] but also to the implementation of quantum gates themselves [33,34].

Throughout the extensive research conducted, considerable attention has also been devoted to the models of QWs themselves. Initially, QW models were introduced in the form of *coined* QWs, wherein the time evolution of walkers is considered through the utilization of unitary matrices called coin matrices, analogous to transitions in RWs [9]. According to the No-Go Lemma [35,36], we need the additional subspace, namely the coin space to each vertex, to construct a non-trivial unitary time evolution on the cycle, introduced later. This model remains one of the most intuitive models and is widely studied or applied even in contemporary times. On the other hand, depending on the research interests, some literature also explores QW models that do not incorporate coins [37,38,39], which represent a generalization of quantum cellular automata [35]. The coinless QW models seem to achieve reduced computational costs, but it is shown that such QW models are unitarily equivalent to coined QW models [40]. In this paper, we focus on using the coined QW models because adopting the coined QWs is reasonable when conducting a comparison with RWs. In the following, when referring to QWs, it indicates the *coined* QWs.

This paper proposes new solution schemes for multi-armed bandit (MAB) problems [41] using RWs and QWs. In the MAB problems, we consider a situation where there are multiple slot machines in an environment, each gives a reward with a probability allocated to it, and an agent iterates the selection of slot machines and probabilistic gain of the rewards and tries to maximize the total reward. Initially, the agent has no information about the probability of giving rewards, especially which slot machine has the maximum probability, which we call the best slot machine. Thus, it is required to accumulate such information through a certain number of selections, an action which we call *exploration*. On the other hand, too much exploration will use up the opportunities for selecting the better slot machines that have already been found; that is, it is also necessary to spend some rounds to bet on slot machines that are reliable based on the information obtained, which we call *exploitation*. The difficulty of MAB problems occurs under the balance between these two operations, known as the *exploration–exploitation trade-off* [42].

One of the purposes of this study is to show that, by utilizing QWs, we can construct an algorithm to solve the MAB problem. This algorithm can outperform models implemented by RWs in terms of the total rewards under some settings. Our idea to realize this is to address this dilemma derived from the exploration–exploitation trade-off by utilizing a unique property of QWs, i.e., the coexistence of linear spreading and localization. More precisely, we combine exploration with linear spreading, and exploitation with localization, as shown in Figure 1. By utilizing linear spreading, we intend to cover the whole environment and prevent us from missing some slot machines. In addition, by applying localization, we intend to mark slot machines that should be recommended with a high probability distribution. This paper introduces a QW-based algorithm for MAB problems, which realizes these combinations. Our study focuses on three-state site-dependent QWs on a cycle. The behavior of these walks corresponds to that of lazy random walks, with walkers moving clockwise, anti-clockwise, or staying in place in superposition. Giving three states to QWs enables us to obtain a high existence probability at the initial position of a QWer. In addition, site-dependent coin matrices make it possible to trap or dam the QWer on certain vertices. By taking advantage of these, we attempt to bring about a high probability on a vertex whose slot machine should be recommended. To facilitate a clear comparison, we also construct an RW-based model wherein lazy RWs occur on cycles, and the transition probabilities depend on the position from which the walkers depart. While the QW-based model possesses the coexistence of linear spreading and localization, the RW-one does not; while the results depend on the specifics of the MAB problem, our study reveals that, for certain settings, the different properties of QWs and RWs also lead to a significant difference in total rewards between both algorithms.

The rest of this paper is organized as follows. First, in Section 2, we present an algorithm for MAB problems based on the RW, which is more intuitive than the QW-based one. Then, in Section 3, we introduce a system of discrete-time quantum walks on a cycle and the QW-based algorithm for MAB problems. In Section 4, we show some results for numerical simulations of the RW- and QW-based models and compare the performance between the two models. Section 5 concludes this paper and discusses the future possibilities of our work.

## 2. Random-Walk-Based Model for MAB Problem

This section presents an MAB algorithm implemented using a discrete-time random walk (RW) on cycles. The RW model presented in this paper describes the walkers that can stay at the same position, which we often call a *lazy random walk*. First, we present the mathematical system of the lazy RW, and then we construct the MAB algorithm based on it.

### 2.1. Random Walk on Cycles

Assume that cycle $C_{N}$ is composed of *N* vertices and edges. Here vertices are labeled by set $V_{N}:=\left\{0,\;1,\;\cdots ,\;N-1\right\}$, and the label is ordered clockwise. Thus, the set of edges is given by $E_{N}:=\left\{\left\{x,\;x+1\right\}|x\in V_{N}\right\}$, where one applies addition and subtraction modulo *N* to $V_{N}$; i.e., (N−1)+1≡0 and 0−1≡N−1. In other words, $V_{N}$ is isometric to Z/NZ.

We assume that the position of a walker is determined as follows:A walker initially exists at position $s\in V_{N}$.At each time step, a walker at position *x* moves one unit clockwise with probability q(x), moves one unit anti-clockwise with probability q(x) or stays at the current position with probability 1−2q(x).

Here the probabilities of moving clockwise and anti-clockwise are equal to each other in this paper. This is due to correspondence with the setting of the QW presented later; therein, the setting of the initial state of the QW gives symmetric probability distributions when the coin matrix is homogeneous for the space. Note that q(x) should satisfy 0≤q(x)≤1/2 under this condition, and this setting is equivalent to the simple RW on cycles in the case of q(x)=1/2 for all $x\in V_{N}$.

Such an RW is mathematically constructed as follows. For $N_{0}=\left\{0,\;1,\;\cdots \right\}$, let ${\left\{X_{t}\right\}}_{t\in N_{0}}$ be the sequence of random variables that represent the position of a walker at time step *t*. The next position $X_{t+1}$ depends on the current one $X_{t}$, and the conditional probability is determined as follows: (1)$$\begin{array}{cc} \; & P\left(X_{t+1}=x+1\;|\;X_{t}=x\right)=P\left(X_{t+1}=x-1\;|\;X_{t}=x\right)=q\left(x\right), \end{array}$$(2)$$\begin{array}{cc} \; & P\left(X_{t+1}=x\;|\;X_{t}=x\right)=1-2q\left(x\right) \end{array}$$
with $x\in V_{N}$. We recall that $V_{N}≃Z/NZ$; the equation above includes
(3)$$\begin{array}{cc} \; & P\left(X_{t+1}=N-1\;|\;X_{t}=0\right)=q\left(0\right), \end{array}$$
(4)$$\begin{array}{cc} \; & P\left(X_{t+1}=0\;|\;X_{t}=N-1\right)=q\left(N-1\right). \end{array}$$
Here we denote the probability that a walker is at position *x* at time step *t* by $\nu^{\left(t\right)}\left(x\right)$:(5)$$\begin{array}{c} \nu^{\left(t\right)}\left(x\right)=P\left(X_{t}=x\right). \end{array}$$
Then, the relation
(6)$$\begin{array}{cc} \; & \nu^{\left(0\right)}\left(x\right)=\delta_{s}\left(x\right) \end{array}$$
holds, where $\delta_{x^{\prime }}\left(x\right)$ is the delta function: for $x^{\prime }\in V_{N}$,
(7)$$\begin{array}{c} \delta_{x^{\prime }}\left(x\right)=\left\{\begin{array}{cc} 1 & \left(x=x^{\prime }\right)\\ 0 & \left(\mathrm{otherwise}\right) \end{array}\right.. \end{array}$$
Moreover, by Equations (Equation 1) and (Equation 1), $\nu^{\left(t\right)}\left(x\right)$ varies as follows:(8)$$\begin{array}{cc} \; & \nu^{\left(t+1\right)}\left(x\right)=q\left(x+1\right)\nu^{\left(t\right)}\left(x+1\right)+\left(1-2q\left(x\right)\right)\nu^{\left(t\right)}\left(x\right)+q\left(x-1\right)\nu^{\left(t\right)}\left(x-1\right). \end{array}$$

### 2.2. Random-Walk-Based Algorithm

We consider an *N*-armed bandit problem with cycle $C_{N}$; each vertex $x\in V_{N}$ is given a slot machine that gives a reward with the probability p(x). In the following, each slot machine is identified by the same label as the corresponding slot machine; for example, we call the slot machine on the vertex *x* slot machine *x*. In addition, we call probability p(x) the *success probability of slot machine x*. Moreover, we denote the slot machine with the best success probability in $V_{N}$ by $x^{*}$; that is,
(9)$$\begin{array}{c} x^{*}=\underset{x\in V_{N}}{arg max}p\left(x\right), \end{array}$$
and we call it the *best slot machine*.

The principle consists of the following four steps: **[STEP 0]** initializing the quantum walk settings, **[STEP 1]** running random walks, **[STEP 2]** playing the selected slot machine, and **[STEP 3]** updating the quantum walk settings. After finishing **[STEP 3]**, the process returns to **[STEP 1]**. We call the series of the last three steps (**[STEP 1–3]**, shown in Figure 2) a *decision*, and decisions are iterated *J* times over a run. Here, we use the following notations:
$s_{j}\in V_{N}$: Initial position of random walk in the *j*-th decision.$q_{j}\left(x\right)\in \left[0,\;1/2\right]$: Clockwise-transition probability and anti-clockwise-transition probability in the *j*-th decision.${\hat{x}}_{j}\in V_{N}$: Vertex (slot machine) measured in the *j*-th decision.${\hat{r}}_{j}\in \left\{0,\;1\right\}$: Reward on the *j*-th decision. This value is probabilistically determined by the Bernoulli distribution $\mathrm{Ber}(p\left({\hat{x}}_{j}\right))$; that is,
(10)$$\begin{array}{c} {\hat{r}}_{j}:=\left\{\begin{array}{cc} 1 & (\mathrm{with}\;\mathrm{prob}.\;p({\hat{x}}_{j}))\\ 0 & (\mathrm{with}\;\mathrm{prob}.\;1-p({\hat{x}}_{j})) \end{array}\right.. \end{array}$$$H_{j}\left(x\right)$: Number of decisions where the slot machine *x* is selected until the *j*-th decision.$L_{j}\left(x\right)$: Number of decisions where the slot machine *x* gives the reward until the *j*-th decision.${\hat{p}}_{j}\left(x\right)$: Empirical probability that the slot machine *x* gives the reward on the *j*-th decision:
(11)$$\begin{array}{c} {\hat{p}}_{j}\left(x\right)=\left\{\begin{array}{cc} \frac{\;L_{j}\left(x\right)\;}{\;H_{j}\left(x\right)\;} & \left(H_{j}\left(x\right)\neq 0\right)\\ 0 & \left(H_{j}\left(x\right)=0\right) \end{array}\right.. \end{array}$$


**[STEP 0] RW-setting initialization**


For the first decision, the settings of the random walk are determined as follows:Initial position $s_{1}$: Probabilistically determined by the uniform distribution on $V_{N}$.Transition probability: $q_{1}\left(x\right)=q^{\circ }\in \left[0,\;1/2\right]$ for all $x\in V_{N}$.

After finishing this step, the process iterates the following three steps. 


**[STEP 1] Random walk**


Random walks are run over *T* time steps with the initial position $s_{j}$ and transition probability $q_{j}\left(x\right)$, and the value ${\hat{x}}_{j}\in V_{N}$ is obtained following probability distribution $\nu_{j}^{\left(T\right)}\left(x\right)$. 


**[STEP 2] Slot machine play**


The slot machine ${\hat{x}}_{j}\in V_{N}$ obtained at **[STEP 1]** is played. Then, the reward $\left({\hat{r}}_{j}=1\right)$ is obtained with probability $p({\hat{x}}_{j})$.

Here *H*- and *L*-values are updated. First, the *H*-value on ${\hat{x}}_{j}$ is incremented:(12)$$\begin{array}{c} H_{j}\left({\hat{x}}_{j}\right)=H_{j-1}\left({\hat{x}}_{j}\right)+1. \end{array}$$
If ${\hat{r}}_{j}=1$, the *L*-value on ${\hat{x}}_{j}$ is also incremented (otherwise, the value is maintained):(13)$$\begin{array}{c} \begin{array}{c} L_{j}\left({\hat{x}}_{j}\right)=\left\{\begin{array}{cc} L_{j-1}\left({\hat{x}}_{j}\right)+1 & (\mathrm{with}\;\mathrm{prob}.\;p({\hat{x}}_{j}))\\ L_{j-1}\left({\hat{x}}_{j}\right) & (\mathrm{with}\;\mathrm{prob}.\;1-p({\hat{x}}_{j})) \end{array}\right.. \end{array} \end{array}$$
For $x\neq {\hat{x}}_{j}$, the *H*- and *L*-values are maintained: (14)$$\begin{array}{cc} H_{j}\left(x\right) & =H_{j-1}\left(x\right), \end{array}$$(15)$$\begin{array}{cc} L_{j}\left(x\right) & =L_{j-1}\left(x\right). \end{array}$$
Based on that, ${\hat{p}}_{j}\left(x\right)$s are updated. 


**[STEP 3] RW-setting adjustment**


Using the new ${\hat{p}}_{j}\left(x\right)$s, the settings of quantum walks are updated for the next decision. The new initial position is defined as
(16)$$\begin{array}{c} s_{j+1}=\underset{x\in V_{N}}{arg max}{\hat{p}}_{j}\left(x\right). \end{array}$$

Moreover, the new transition probabilities are determined as
(17)$$\begin{array}{c} q_{j+1}\left(x\right)=q^{\circ }\exp\left(-a\cdot {\hat{p}}_{j}{\left(x\right)}^{b}\right) \end{array}$$
where a,b≥1, and $q^{\circ }$ are defined in **[STEP 0]**. Note that the *q*-value monotonically decreases about the empirical success probability; that is, if ${\hat{p}}_{j}\left(x\right)$ is larger, then $q_{j+1}\left(x\right)$ is smaller. By setting the new initial state and *q*-value in these manners, we aim at confining walkers to the desired position while concurrently affording them opportunities to depart when the current decision is uncertain. The parameters *a* and *b* control the strength of the effect by ${\hat{p}}_{j}\left(x\right)$; the details are given in Appendix A.

After this step, the process returns to **[STEP 1]**. 

## 3. Quantum-Walk-Based Model for MAB Problem

This section presents the MAB algorithm implemented by the discrete-time quantum walk (QW) on cycles. The difference between QW and RW lies in whether one handles the quantum superposition of states pertaining to the walker’s positions. Herein, the transition of walkers at each time step is also superposed; that is, it is uncertain even after the time step which transition occurs: moving clockwise, anti-clockwise, or staying in place. We introduce probability amplitude vectors and coin matrices in the QW to describe the quantum superposition and its time evolution.

The QW model employed in our study is a three-state QW on a cycle, which can be naturally reduced to a finite space from the one-dimensional lattice model [5]; see, for example, [18,43,44]. First, we explain the definition of our QW in detail, and then we present the MAB algorithm based on it.

### 3.1. Quantum Walk on Cycles

Assume that cycle $C_{N}$ is constructed in the same manner as in Section 2; that is, it is the graph established by the set of vertices $V_{N}:=\left\{0,\;1,\;\cdots ,\;N-1\right\}$ and edges $E_{N}:=\left\{\left\{x,\;x+1\right\}|x\in V_{N}\right\}$, where one applies addition and subtraction modulo *N* to $V_{N}$.

The space of the probability amplitude vectors driving our QW is defined in a compound Hilbert space consisting of the position Hilbert space $H_{P}$ and the coin Hilbert space $H_{C}$. The position Hilbert space $H_{P}$ is spanned by the unit vectors corresponding to the vertices on $C_{N}$; i.e., $H_{P}=\mathrm{span}\left\{|x\rangle \;|\;x\in V_{N}\right\}$. Here we require them to be mutually orthogonal, which is equivalent to satisfying the relation $\langle y|x\rangle =\delta_{y}\left(x\right)$ for any $x,\;y\in V_{N}$, where $\delta_{y}$ is the delta function defined by Equation (Equation 7). Then, $H_{P}≃C^{N}$ holds.

The coin Hilbert space $H_{C}$ pertains to the internal state of walkers. In this model, we assume that there exist three internal states: clockwise (+), anti-clockwise (−), and staying (O). We define the three-dimensional unit vectors corresponding to them as $|-\rangle ={\left[1\;\;\;0\;\;\;0\right]}^{T}$, $|O\rangle ={\left[0\;\;\;1\;\;\;0\right]}^{T}$, and $|+\rangle ={\left[0\;\;\;0\;\;\;1\right]}^{T}$, where a superscript T on a matrix represents its transpose, and construct the coin Hilbert space as $H_{C}=\mathrm{span}\left\{|-\rangle ,\;|O\rangle ,\;|+\rangle \right\}$. Here you see that $H_{C}=C^{3}$. Based on $H_{P}$ and $H_{C}$, the whole system is described by
(18)$$\begin{array}{c} H_{PC}=H_{P}⊗H_{C}=\mathrm{span}\left\{|x\rangle ⊗|\varepsilon \rangle \;|\;x\in V_{N},\;\varepsilon \in \left\{\pm ,\;O\right\}\right\}. \end{array}$$

Then the total state of our QW at time $t\in N_{0}$ is represented as follows: there exists $|\psi^{\left(t\right)}\left(x\right)\rangle \in C^{3}$ for each $x\in V_{N}$ such that
(19)$$\begin{array}{c} |\Psi^{\left(t\right)}\rangle =\sum_{x\in V_{N}}|x\rangle ⊗|\psi^{\left(t\right)}\left(x\right)\rangle \in H_{PC}. \end{array}$$

Here, $t\in N_{0}$ represents time step of QWs, and $|\psi^{\left(t\right)}\left(x\right)\rangle \in C^{3}$ is called the probability amplitude vector at position $x\in V_{N}$ at run time *t*. We set the initial state as
(20)$$\begin{array}{c} |\Psi^{\left(0\right)}\rangle =|\Phi \rangle :=|s\rangle ⊗|\varphi \rangle , \end{array}$$
where $s\in V_{N}$, and $|\varphi \rangle \in C^{3}$ is a constant vector with ∥φ∥=1. In this paper, we fix |φ〉 to |O〉, realizing a symmetric probability distribution about the initial position when the coin matrix defined later is homogeneous for positions.

Now, we introduce the time evolution of $|\Psi^{\left(t\right)}\rangle$ by
(21)$$\begin{array}{c} |\Psi^{\left(t+1\right)}\rangle =U|\Psi^{\left(t\right)}\rangle . \end{array}$$
Here *U* is the unitary operator, referred to as the time evolution operator, and is composed of shift operator *S* and coin operator *C*: (22)$$\begin{array}{c} U=SC, \end{array}$$
and *S* and *C* are given by
(23)$$\begin{array}{c} S=S^{†}⊗|-\rangle \;\langle -|+I_{N}⊗|O\rangle \;\langle O|+S⊗|+\rangle \;\langle +| \end{array}$$
and
(24)$$\begin{array}{c} C=\sum_{x\in V_{N}}\left(|x\rangle \;\langle x|⊗C(x)\right). \end{array}$$
Here *S* is defined as
(25)$$\begin{array}{c} S=\sum_{x\in V_{N}}|x+1\rangle \;\langle x| \end{array}$$
and represents the clockwise transition, and then
(26)$$\begin{array}{c} S^{†}=\sum_{x\in V_{N}}|x-1\rangle \;\langle x| \end{array}$$
indicates the anti-clockwise transition. The identity matrix $I_{N}$ corresponds to staying in place. Here, note that N≡0 on $V_{N}\left(≃Z/NZ\right)$; that is, for example, in the case of N=4,
(27)$$\begin{array}{c} S=|1\rangle \;\langle 0|+|2\rangle \;\langle 1|+|3\rangle \;\langle 2|+|0\rangle \;\langle 3| \end{array}$$
and
(28)$$\begin{array}{c} S^{†}=|3\rangle \;\langle 0|+|0\rangle \;\langle 1|+|1\rangle \;\langle 2|+|2\rangle \;\langle 3|. \end{array}$$
C(x) is a unitary matrix called a *coin matrix*, which is defined as follows:(29)$$\begin{array}{c} C\left(x\right)=\left[\begin{array}{ccc} -\frac{\;1+\cos\theta (x)\;}{\;2\;} & \frac{\;\sin\theta (x)\;}{\;\sqrt{2}\;} & \frac{\;1-\cos\theta (x)\;}{\;2\;}\\ \frac{\;\sin\theta (x)\;}{\;\sqrt{2}\;} & \cos\theta (x) & \frac{\;\sin\theta (x)\;}{\;\sqrt{2}\;}\\ \frac{\;1-\cos\theta (x)\;}{\;2\;} & \frac{\;\sin\theta (x)\;}{\;\sqrt{2}\;} & -\frac{\;1+\cos\theta (x)\;}{\;2\;} \end{array}\right] \end{array}$$
with θ(x)∈[0,2π) for all $x\in V_{N}$. Note that, in the case of cosθ(x)=−1/3, C(x) is reduced to the Grover matrix, which is important in quantum searching [45].

Let us explain the equivalent expression for the time evolution operator *U*, which is useful to understand the dynamics of our QW: by applying the property of the Kronecker product, we have
(30)$$\begin{array}{cc} \; & U=\sum_{x\in V_{N}}\left(|x-1\rangle \;\langle x|⊗P(x)+|x\rangle \;\langle x|⊗R(x)+|x+1\rangle \;\langle x|⊗Q(x)\right), \end{array}$$
where
(31)$$\begin{array}{cc} P(x) & =|-\rangle \;\langle -|C\left(x\right)=\left[\begin{array}{ccc} -\frac{\;1+\cos\theta (x)\;}{\;2\;} & \frac{\;\sin\theta (x)\;}{\;\sqrt{2}\;} & \frac{\;1-\cos\theta (x)\;}{\;2\;}\\ 0 & 0 & 0\\ 0 & 0 & 0 \end{array}\right], \end{array}$$
(32)$$\begin{array}{cc} Q(x) & =|+\rangle \;\langle +|C\left(x\right)=\left[\begin{array}{ccc} 0 & 0 & 0\\ 0 & 0 & 0\\ \frac{\;1-\cos\theta (x)\;}{\;2\;} & \frac{\;\sin\theta (x)\;}{\;\sqrt{2}\;} & -\frac{\;1+\cos\theta (x)\;}{\;2\;} \end{array}\right], \end{array}$$
(33)$$\begin{array}{cc} R(x) & =|O\rangle \;\langle O|C\left(x\right)=\left[\begin{array}{ccc} 0 & 0 & 0\\ \frac{\;\sin\theta (x)\;}{\;\sqrt{2}\;} & \cos\theta (x) & \frac{\;\sin\theta (x)\;}{\;\sqrt{2}\;}\\ 0 & 0 & 0 \end{array}\right]. \end{array}$$
The matrices P(x), Q(x), and R(x) are considered to be the decomposition elements of C(x); that is, the relation P(x)+Q(x)+R(x)=C(x) holds. They describe the matrix-valued weight of a clockwise transition, an anti-clockwise transition, and staying in place, respectively, corresponding to the transition probabilities of the RW, as shown in Figure 3.

By Equations (Equation 21) and (Equation 30), we have
(34)$$\begin{array}{cc} \; & |\psi^{\left(t+1\right)}\left(x\right)\rangle =P(x+1)|\psi^{\left(t\right)}\left(x+1\right)\rangle +R(x)|\psi^{\left(t\right)}\left(x\right)\rangle +Q(x-1)|\psi^{\left(t\right)}\left(x-1\right)\rangle . \end{array}$$
Moreover, from the initial state (Equation 20), there exists a 2-dimensional matrix $\Xi^{\left(t\right)}\left(x\right)$ such that
(35)$$\begin{array}{c} |\psi^{\left(t\right)}\left(x\right)\rangle =\Xi^{\left(t\right)}\left(x\right)|\varphi \rangle . \end{array}$$
Here $\Xi^{\left(t\right)}\left(x\right)$ describes the weight of all the possible paths from the origin to the position *x* at run time *t*. From Equation (Equation 34), the following relation holds:(36)$$\begin{array}{c} \Xi^{\left(t+1\right)}\left(x\right)=P\left(x+1\right)\Xi^{\left(t\right)}\left(x+1\right)+R\left(x\right)\Xi^{\left(t\right)}\left(x\right)+Q\left(x-1\right)\Xi^{\left(t\right)}\left(x-1\right). \end{array}$$

Finally, the measurement probability of the particle at position *x* at run time *t*, denoted by $\mu^{\left(t\right)}\left(x\right)$, is given by
(37)$$\begin{array}{c} \mu^{\left(t\right)}\left(x\right):={∥\psi^{\left(t\right)}\left(x\right)∥}^{2}. \end{array}$$
Setting random variable $X_{t}$ following the distribution $\mu^{\left(t\right)}$, we call $X_{t}$ the position of a QWer at time *t*. This definition is based on the Born probability interpretation in quantum mechanics. Note that for any $t\in N_{0}$, the following is satisfied:(38)$$\begin{array}{c} \sum_{x\in V_{N}}\mu^{\left(t\right)}\left(x\right)=\sum_{x\in V_{N}}{∥\psi^{\left(t\right)}\left(x\right)∥}^{2}=1. \end{array}$$

### 3.2. Quantum-Walk-Based Algorithm

We consider an *N*-armed bandit problem with cycle $C_{N}$; each vertex $x\in V_{N}$ is given a slot machine that gives a reward with the probability p(x), identically to the RW-based model in Section 3.

The principle is also similar to the RW-based model: First, the QW settings are initialized (**[STEP 0]**), and then decisions are iterated (**[STEP 1–3]**) *J* times over a run. As shown in Figure 4, the QW-based model controls the coin matrix C(x) by adjusting the value of the parameter θ(x) instead of the transition probability q(x) in the RW-based model. Here we use the following notations:$|\Phi_{j}\rangle \in H_{PC}$: Initial state of quantum walk on the *j*-th decision.$s_{j}\in V_{N}$: Initial position of quantum walk on the *j*-th decision.$\theta_{j}\left(x\right)\in \left[0,\;2\pi \right)$: Parameter of Equation (Equation 24) on vertex *x* on the *j*-th decision; then the coin matrix there is C(x).${\hat{x}}_{j}\in V_{N}$: Vertex (slot machine) measured on the *j*-th decision.${\hat{r}}_{j}\in \left\{0,\;1\right\}$: Reward on the *j*-th decision, which follows the Bernoulli distribution $\mathrm{Ber}(p\left({\hat{x}}_{j}\right))$:
(39)$$\begin{array}{c} {\hat{r}}_{j}:=\left\{\begin{array}{cc} 1 & (\mathrm{with}\;\mathrm{prob}.\;p({\hat{x}}_{j}))\\ 0 & (\mathrm{with}\;\mathrm{prob}.\;1-p({\hat{x}}_{j})) \end{array}\right.. \end{array}$$$H_{j}\left(x\right)$: Number of decisions in which slot machine *x* is selected until the *j*-th decision.$L_{j}\left(x\right)$: Number of decisions in which slot machine *x* gives the reward until the *j*-th decision.${\hat{p}}_{j}\left(x\right)$: Empirical probability that slot machine *x* gives the reward on the *j*-th decision:
(40)$$\begin{array}{c} {\hat{p}}_{j}\left(x\right)=\left\{\begin{array}{cc} \frac{\;L_{j}\left(x\right)\;}{\;H_{j}\left(x\right)\;} & \left(H_{j}\left(x\right)\neq 0\right)\\ 0 & \left(H_{j}\left(x\right)=0\right) \end{array}\right.. \end{array}$$


**[STEP 0] QW-setting initialization**


For the first decision, the settings of the quantum walk are determined as follows:Initial state: $|\Phi_{1}\rangle =|s_{1}\rangle ⊗|O\rangle$. Here the initial position $s_{1}$ is probabilistically determined by the uniform distribution on $V_{N}$.Parameter of coin matrices: $\theta_{1}\left(x\right)=\theta^{\circ }\in \left[0,\;2\pi \right)$ for all $x\in V_{N}$.

After finishing this step, the run iterates the following three steps. 


**[STEP 1] Quantum walk**


Quantum walks are run over *T* time steps with the initial position $s_{j}$ and the parameter $\theta_{j}\left(x\right)$. After running *T* steps of time evolution, the QWer is measured to obtain the value ${\hat{x}}_{j}\in V_{N}$ following probability distribution $\mu_{j}^{\left(T\right)}\left(x\right)$. 


**[STEP 2] Slot machine play**


The slot machine ${\hat{x}}_{j}\in V_{N}$ obtained at **[STEP 1]** is played. Then, the reward (${\hat{r}}_{j}=1$) is obtained with probability $p({\hat{x}}_{j})$.

Here *H*- and *L*-values are updated. First, the *H*-value on ${\hat{x}}_{j}$ is incremented:(41)$$\begin{array}{c} H_{j}\left({\hat{x}}_{j}\right)=H_{j-1}\left({\hat{x}}_{j}\right)+1. \end{array}$$
If ${\hat{r}}_{j}=1$, the *L*-value on ${\hat{x}}_{j}$ is also incremented (otherwise, the value is maintained):(42)$$\begin{array}{c} \begin{array}{c} L_{j}\left({\hat{x}}_{j}\right)=\left\{\begin{array}{cc} L_{j-1}\left({\hat{x}}_{j}\right)+1 & (\mathrm{with}\;\mathrm{prob}.\;p({\hat{x}}_{j}))\\ L_{j-1}\left({\hat{x}}_{j}\right) & (\mathrm{with}\;\mathrm{prob}.\;1-p({\hat{x}}_{j})) \end{array}\right.. \end{array} \end{array}$$
For $x\neq {\hat{x}}_{j}$, the *H*- and *L*-values are maintained: (43)$$\begin{array}{cc} H_{j}\left(x\right) & =H_{j-1}\left(x\right), \end{array}$$(44)$$\begin{array}{cc} L_{j}\left(x\right) & =L_{j-1}\left(x\right). \end{array}$$
Based on that, the ${\hat{p}}_{j}\left(x\right)$ values are updated. 


**[STEP 3] QW-setting adjustment**


Using the new ${\hat{p}}_{j}\left(x\right)$s, the settings of quantum walks for the next decision are updated. The new initial state is defined as
(45)$$\begin{array}{c} |\Phi_{j+1}\rangle =|s_{j}\rangle ⊗|O\rangle , \end{array}$$
where $s_{j}$ is the provisionally best machine:(46)$$\begin{array}{c} s_{j}=\underset{x\in V_{N}}{arg max}{\hat{p}}_{j}\left(x\right). \end{array}$$

Moreover, the new parameters of the coin matrices are determined as
(47)$$\begin{array}{c} \theta_{j+1}\left(x\right)=\theta^{\circ }\exp\left(-a\cdot {\hat{p}}_{j}{\left(x\right)}^{b}\right) \end{array}$$
where a,b≥1, and $\theta^{\circ }$ are defined in **[STEP 0]**. Note that the θ-value is defined similarly to the *q*-value in the RW-based model; that is, if ${\hat{p}}_{j}\left(x\right)$ is larger, then $\theta_{j+1}\left(x\right)$ is smaller. When the θ-value at a certain position $x_{L}\in V_{N}$ is updated, the difference between $C(x_{L})$ and C(x) with $x=x_{L}\pm 1$ emerges, wherein $x_{L}$ is often called a *defect*. If a defect exists at the initial position, the coin matrix, depending on the θ-value, controls the strength of localization there. Incidentally, when ${\hat{p}}_{j}\left(x\right)$ is large, $\theta_{j+1}\left(x\right)$ can be almost 0. If θ(x) in Equation (Equation 24) is exactly 0, then C(x) is the identity matrix. This means that, if the initial position of the walker has a coin matrix with θ(x)=0, the walker is completely trapped there because the internal state is set to be |O〉. Thus, a larger empirical success probability indicates a strong localization if it is provisionally best. However, if ${\hat{p}}_{j}\left(x\right)$ is not large, this phenomenon is relaxed. In short, this θ-value plays a role corresponding to the *q*-value in the RW-based model; that is, it confines walkers to the desired position while concurrently affording them opportunities to depart when the current decision is uncertain. Regarding the analysis of θ-values, also see Appendix A.

After this step, the process returns to **[STEP 1]**. 

## 4. Numerical Simulations

In this section, we give and compare simulation results for the RW- and QW-based models. Assume that the RW- and QW-based models are run in parallel *K* times, respectively, and each run is labeled by the set {1,2,⋯,K}. We indicate that the parameters are in the *k*-th run by a subscript next to the number of iterations; for example, the reward in the *j*-th decision in the *k*-th run is denoted by ${\hat{r}}_{j,\;k}$.

As figures-of-merit, we define quantities M(j), ρ(j), and CDR(j): (48)$$\begin{array}{cc} M(j) & :=\frac{\;1\;}{\;K\;}\sum_{k=1}^{K}\sum_{\ell =1}^{j}{\hat{r}}_{\ell ,\;k}, \end{array}$$(49)$$\begin{array}{cc} \rho (j) & :=\frac{\;1\;}{\;K\;}\sum_{k=1}^{K}\sum_{\ell =1}^{j}\left(p\left(x^{*}\right)-p\left({\hat{x}}_{\ell ,\;k}\right)\right), \end{array}$$(50)$$\begin{array}{cc} \mathrm{CDR}(j) & :=\frac{\;1\;}{\;K\;}\sum_{k=1}^{K}\delta_{x^{*}}\left({\hat{x}}_{j,\;k}\right). \end{array}$$
M(j) indicates the mean of *total rewards* until the *j*-th decision over *K* runs. The aim of the proposed models is to make M(j) as large as possible. ρ(j) is the mean of *cumulative regret* until the *j*-th decision over *K* runs. The cumulative regret is equal to the difference in expectations of total reward between the case where only the best machine is selected until the *j*-th decision and that of actual selections until then. CDR(j) is the *correct decision rate* of the *j*-th decision, which is the ratio of the number of runs in selecting the best slot machine to the total number of runs *K*. Herein, $\delta_{y}$ for $y\in V_{N}$ is the delta function defined by Equation (Equation 7).

The parameter values used for this series of simulations are summarized in Table 1. The success probabilities of slot machines are given as shown in Figure 5; that is,
(51)$$\begin{array}{c} p\left(x\right)=\left\{\begin{array}{cc} 0.9 & \left(x=14\right)\\ 0.1 & \left(x=15\right)\\ 0.7 & \left(x:\;\mathrm{even}\;\mathrm{except}\;\mathrm{for}\;14\right)\\ 0.5 & \left(x:\;\mathrm{odd}\;\mathrm{except}\;\mathrm{for}\;15\right) \end{array}\right.. \end{array}$$

Herein, the best slot machine is $x^{*}=14$. Recall that the agent cannot directly access all information regarding the success probabilities of the slot machines above. The tuples for the QW- and RW-based models are selected as one of the best performers in a certain range of parameters on each model. The details about parameter-dependencies of both models are found in Appendix A.

The blue and orange curves in Figure 6a–c demonstrate the performances of RW- and QW-based models as the variations of the mean of total reward M(j), the cumulative regret ρ(j), and the maximum value of CDR over the number *T* of time steps of walks for single decision-making, respectively. The total reward and the cumulative regret are taken for the final decision J=5000. The maximum value of CDR is taken over *J* decisions; that is,
(52)$$\begin{array}{c} \max\left(\mathrm{CDR}\right):=\max_{j=1,\;\cdots ,\;J}\mathrm{CDR}\left(j\right). \end{array}$$

For T≥4, we observe that M(5000) and max(CDR) of the QW-based model are larger than those of the RW-based model. On the other hand, for the cumulative regret ρ(5000), the value for the QW-based model is lower than that for the RW-based model. Both results indicate that the performance of the QW-based model is superior to that of the RW-based model. You see the particular difference in the growth of M(5000) and max(CDR) over the variation of *T* between the QW- and RW-models; the gradient of the orange curves (QW) in the range of 2≤T≤8 in M(5000) and max(CDR) is larger than that of the blue curve (RW). In addition, the QW-based model has higher suprema of M(5000) and max(CDR) than the RW-based model. Similar discussions are also made for regret ρ(5000). In both the RW- and QW-models ρ(5000) decreases over the variation of *T* in the range of 2≤T≤8, but the gradient of the QW-model is steeper than that of the RW-one. Moreover, the QW-based model has a lower infimum than the RW-based model.

These results depend on a variety of choices; in particular, the casino setting and the parameters *a* and *b*. Indeed, the QW-based model shows a faster growth in reward for many of *a* and *b* compared to the RW-based model (see Appendix A). However, there are also settings where this relationship is reversed. We have also observed that in the limit of very large *a* and *b*, when fine-tuned to a specific casino setting, the performance of both RW and QW can grow even faster than the results shown in Figure 6. We speculate that linear spreading and localization may lose their advantage for certain settings.

You can see the contribution of linear spreading and localization from the behavior of variation of decision-making and probability distributions for making a decision, which is particularly apparent for smaller *T*. Figure 7 and Figure 8, respectively, show the precise performances of runs of the RW- and QW-based models whose resultant total rewards M(J) were almost equal to the average value. Herein, the number of time steps *T* is set to be 8 for each model, and the other parameters are set as in Table 1. Figure 7a,b indicate the relationship between the decision *j* and the selected slot machine ${\hat{x}}_{j}$ for the RW- and QW-based models, respectively. From this figure, you can see that the decision-making in the QW-based model almost converges to $x=x^{*}$ near j=1200, while that in the RW-based model does so near j=1400. This means that exploration in the QW-based model is more successful than that in the RW-based model. Linear spreading makes the probability distribution of QWs wider, whose variance is larger than that of RWs, which results in faster exploration of the QW-based model. Moreover, the behavior of the QW-based model after finding $x=x^{*}$ is more stable than that of the RW-based model, which indicates that the QW-based model also realizes more effective exploitation than the RW-based one for this set of parameters.

These behaviors are interpreted by the variations of the probability distributions of the RW($\nu_{j}^{\left(T\right)}$)- and QW($\mu_{j}^{\left(T\right)}$)-based models over decision *j* shown in Figure 8a,b, respectively. You see that, for smaller *j*, the probability of QW is more widely distributed than that of RW, although the values are quite small except for a certain position. It is important that the probabilities are distributed in a wider range even if they are quite small because it indicates that the agent has more selections, which is crucial to realize exploration. As a result, the QW-based model can obtain a high measurement probability of the best slot machine at j=1200 at the latest, while the RW-based model does at j=1400, which corresponds to the convergence of the decision-making shown in Figure 7. After beginning the concentrated investment to $x=x^{*}$, the measurement probability of walkers on the QW is almost 1, while the corresponding probability on the RW is around 0.9, which shows that strong localization occurs on vertex $x^{*}$ after finding the slot machine there. This phenomenon contributes to exploitation; as you see in Figure 7, the QW-based model after finding the best slot machine is much more likely to select it than the RW-based one, and we can understand that it comes from the difference in probability distributions between the two models.

## 5. Conclusions and Discussion

This paper has proposed new solution schemes for multi-armed bandit (MAB) problems using random walks (RWs) and quantum walks (QWs). We demonstrated that we could find parameter regimes where the QW-based model performs better than the RW-based model by addressing the exploration–exploitation dilemma by utilizing a unique property of QWs, i.e., the coexistence of linear spreading and localization. Our idea was to combine exploration with linear spreading and exploitation with localization. By utilizing linear spreading, we expect the QWs to cover the whole environment to prevent it from missing some slot machines. In addition, by applying localization, we expect the QWs to identify the slot machine that should be recommended with a high probability distribution. Indeed, we showed that, under some settings, linear spreading contributes to exploring the environment and quickly finding the best slot machine, and localization contributes to exploiting the best slot machine more frequently.

The positive results obtained in this study open the possibility for further extensions of this approach. First, can we apply this algorithm to the case of multi-agent systems such as competitive or adversarial bandit problems [46,47,48] with some revision? Especially for the QW-based model, we will examine its application to the use of coin matrices implemented by multiple registers or to drive walkers on a torus. Moreover, there are possibilities for constructing application models in the single-agent case. For example, we may construct an evolved version of the QW-based model, including a quantum version of the optimal stopping problem.

Moreover, analyses of our models are also important. The performances obtained in Section 4 should depend on the number of slot machines (i.e., vertices) *N*, the true success probability p(x), and the parameter settings $\left(a,\;b,\;q^{\circ }\right)$ or $\left(a,\;b,\;\theta^{\circ }\right)$. Obtaining theoretical formulae for the figures-of-merit would be desirable, which would make our results more confident. Specifically, it is not immediately evident, given the parameter configurations outlined in this paper, that the QW-based model outperforms the RW-based model in general. Our investigations have focused solely on pairs (a,b) with small values; in the event that larger values are assigned to either *a* or *b*, the RW-based model can exhibit superior performance compared to any QW-based model (assuming the parameters are chosen correctly for a specific casino setting). Indeed, we have confirmed that the RW-based model with tuple $\left(a,\;b,\;q^{\circ }\right)=\left(50,\;20,\;0.5\right)$ performs as well as the QW-based one with tuple $\left(a,\;b,\;\theta^{\circ }\right)=\left(50,\;20,\;29\pi /64\right)$, and in both results, the mean of total reward is around 4320 even in setting the number of time steps *T* to 8 (better than the results for small *a* and *b* as shown in Figure 6). However, sufficiently addressing the parameter dependency and its interplay with the casino settings is a highly complex problem at this stage. We provide our existing analysis in Appendix A.

Several factors contribute to the complexity of a rigid mathematical treatment of this problem, but the one that should be remarked on is the position-dependency of the coin matrices in the QW-based model. Solving QW-based models with site-dependent coins is very difficult, in general. While some studies have addressed this matter for the case where the coin matrix only on the origin differs from the others [6,22] or that the coin matrices are controlled by the trigonometric function whose input is in proportion to the label position [12,14,21], the generalized case remains an open problem. To conduct a thorough analysis of this model, it is necessary to accumulate analytical results for site-dependent quantum walks over an extended period of time.

## Figures and Tables

**Figure 1 entropy-25-00843-f001:**
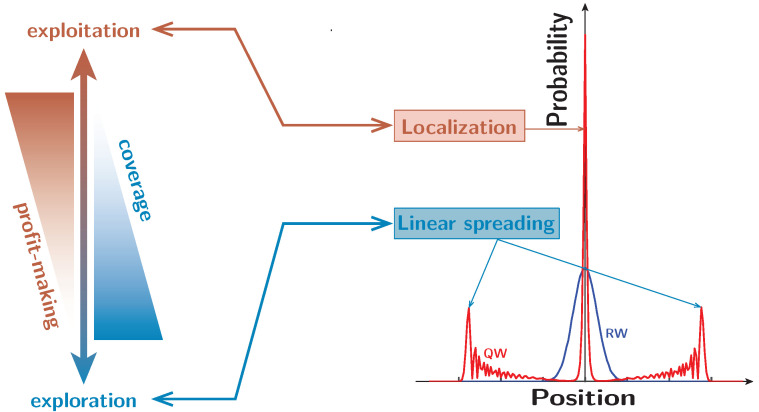
Association between the behaviors of quantum walks (linear spreading and localization) and the operations in MAB problems (exploration and exploitation).

**Figure 2 entropy-25-00843-f002:**
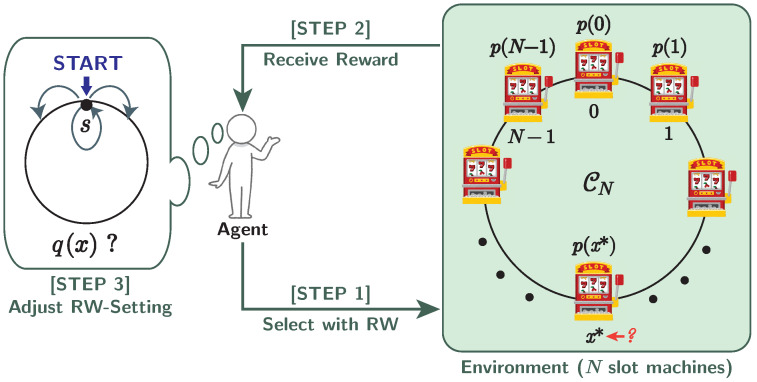
Single decision on the random-walk-based model for MAB problems.

**Figure 3 entropy-25-00843-f003:**
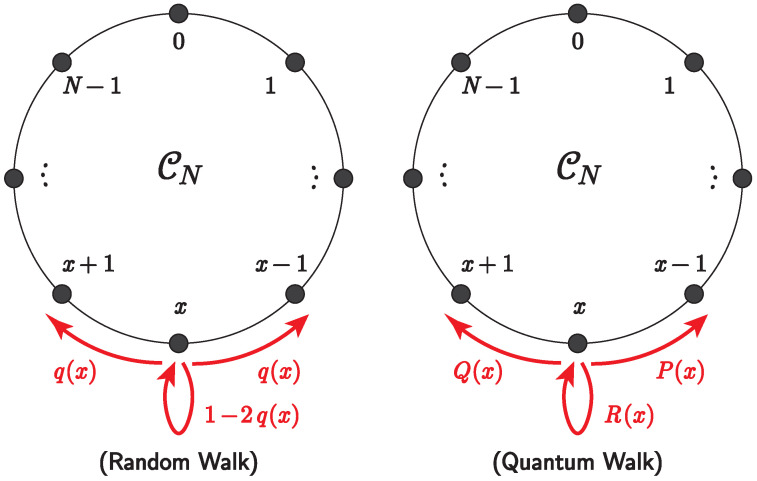
Transition probabilities of RW (**left panel**) and matrix-valued weights of QW (**right panel**).

**Figure 4 entropy-25-00843-f004:**
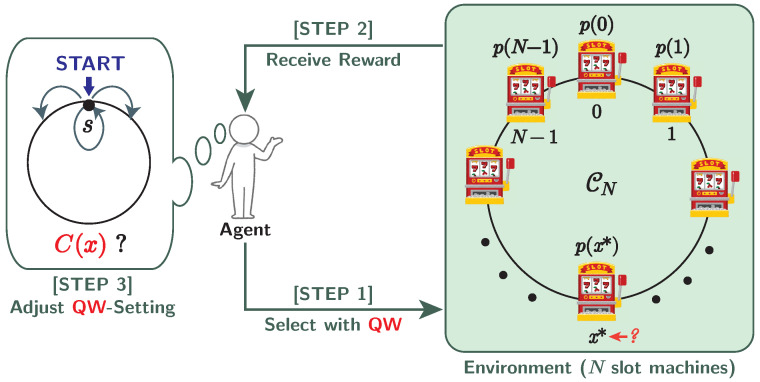
Single decision on the quantum-walk-based model for MAB problems.

**Figure 5 entropy-25-00843-f005:**
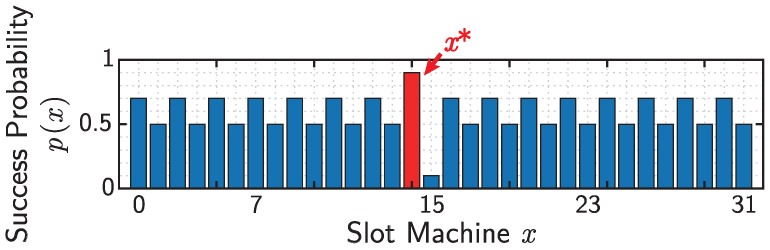
Success probability p(x) for slot machine $x\in V_{N}$. The number of slot machines *N* is set to 32, and the best slot machine is $x^{*}=14$.

**Figure 6 entropy-25-00843-f006:**
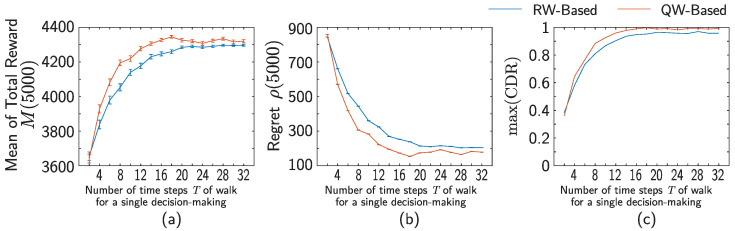
Comparison of (**a**) mean of total reward M(J), (**b**) cumulative regret ρ(J), and (**c**) the maximum value of CDR over the variation of final time step *T* of walks between the RW- and QW-based models. Parameters are determined as shown in Table 1.

**Figure 7 entropy-25-00843-f007:**
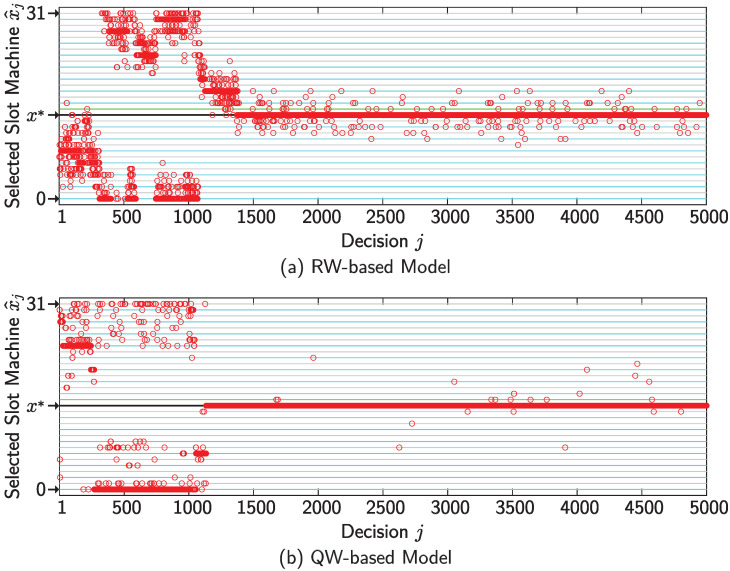
The red markers show the variation of the selected slot machine ${\hat{x}}_{j}$ over decision *j* for single runs of the (**a**) RW- and (**b**) QW-based models. For both settings, the number of time steps *T* is set to 8, and other parameters are determined as in Table 1. Each run is selected as the one whose resultant total rewards M(J) were almost equal to the average value: M(J)=4063 in (**a**), and M(J)=4200 in (**b**). The black, sky blue, gray, and light green lines indicate the slot machines whose success probabilities are 0.9, 0.7, 0.5, and 0.1, respectively.

**Figure 8 entropy-25-00843-f008:**
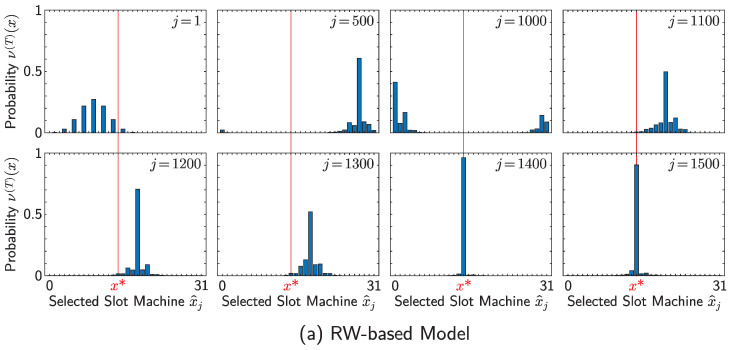
The probability distributions regarding the selected slot machine ${\hat{x}}_{j}$ where walkers exist after *T* steps of walk in the *j*-th decision with j=1,500,1000,1100,1200,1300,1400,1500 for single runs of the (**a**) RW- and (**b**) QW-based models. The settings are the same as in Figure 7; that is, for both settings, the number of time steps *T* is set to 8, other parameters are determined as in Table 1, and each run is selected as the one whose resultant total rewards M(J) were almost equal to the average value: M(J)=4063 in (**a**), and M(J)=4200 in (**b**).

**Table 1 entropy-25-00843-t001:** Parameter values used for numerical simulation of decision making.

Parameter	Symbol	Value
Number of slot machines	*N*	32
Number of runs	*K*	500
Number of decisions for a single run	*J*	5000
Parameters for the QW-based model	$\left(a,\;b,\;\theta^{\circ }\right)$	(5,6,5π/16)
Parameters for the RW-based model	$\left(a,\;b,\;q^{\circ }\right)$	(9,6,0.5)

## Data Availability

The data that support the findings of this study are available from the corresponding author upon reasonable request.

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
