# Peer review of "Bandit Algorithm Driven by a Classical Random Walk and a Quantum Walk"

_entropy, 2023, doi:10.3390/e25060843_

Round 1
Reviewer 1 Report
In this work, the authors study the multi-armed bandit problems using random walks and quantum walks, respectively. The results show that the quantum walk algorithm outperforms the random walk one. The authors introduced two advantages of quantum walks linear spreading and localization.
This work is very interesting. As far as I know, only two papers theoretically investigated the quantum algorithms on multi-armed bandit problems. In this manuscript, the authors really demonstrated the quantum advantages of solving multi-armed bandit problems rather than messy mathematics. In addition, the manuscript is clearly written and well-organized. The only thing I would like to complain about is the parameters a and b, which appear in (24). The authors should put more words on explaining (24) and the parameter setting in the numerical simulations.
Reviewer 2 Report
In the manuscript, an interesting aspect and application of quantum random walks is addressed: so called multi-armed bandit (MAB) problems, where an agent explores an unknown territory to find the best place. There is a tradeoff between exploitation of a given place and exploration (search for the better place). Classically, various models of random walk (RW) are used in these models. Here, the authors propose to use the quantum walks (QW) instead, which have some advantages (faster spreading and localization simultaneously). An agent based on QW is proposed and it is shown that it gives slightly better results, than agents based on RW.
I find the manuscript interesting and recommend it for publication. I suggest the authors to take into account the following comments:
1. Mechanism of the advantage of QW. From Fig. 5, we see that an agent based on QW performs slightly better than an agent based on RW. Do we observe a picture with simultaneous spreading and localization like that on Fig. 1 (the right-hand part) for the particular model of QW considered here? Probably, it is worthwhile to add a plot with actual distributions mu^t(x) and nu^t(x) for QW and RW, respectively.
2. More explanations on the meaning of the parameter theta(x) in Eq. (13) and the decision rule (10) (above line 130) are desired. The corresponding rule (24) (above line 189) for RW is clear: if the empirical success probability is high, then the walk tends not to move from the current node: q(x)=0 corresponds to the case where an agent does not move from the node x. However, if I understand Eqs. (13) and (15) on p.4 correctly, theta(x)=0 means that we keep moving in the same direction or staying: the matrix C is unity (up to signs) and the coin does not change its state. For example, if its state is |->, we move left and the coin does not change its state, i.e., we also move left in the next turn. What is the motivation for such decision?
3. On p.6 the numeration of formulas is reset and the formula above line 95 has again the number (1).
Reviewer 3 Report
Please find attached file.

Round 2
Reviewer 1 Report
I appreciate the authors' effort to make the manuscript crystally clear, and it deserves its publication in Entropy.
Reviewer 2 Report
I'm satisfied with author's reply and corrections in the manuscript and thank the authors for their efforts.
Reviewer 3 Report
Thank you for your detailed reply and I am fully satisfied with the added content.